# The Effect of Roux-en-Y Gastric Bypass on Non-Alcoholic Fatty Liver Disease Fibrosis Assessed by FIB-4 and NFS Scores—An 11.6-Year Follow-Up Study

**DOI:** 10.3390/jcm11164910

**Published:** 2022-08-21

**Authors:** Elfrid Christine Smith Sandvik, Kristin Matre Aasarød, Gjermund Johnsen, Dag Arne Lihaug Hoff, Bård Kulseng, Åsne Ask Hyldmo, Hallvard Græslie, Siren Nymo, Jorunn Sandvik, Reidar Fossmark

**Affiliations:** 1Department of Clinical and Molecular Medicine, Norwegian University of Science and Technology (NTNU), 7491 Trondheim, Norway; 2Department of Gastroenterology and Hepatology, St. Olav’s University Hospital, 7006 Trondheim, Norway; 3Obesity Research Group, Department of Clinical and Molecular Medicine, Faculty of Medicine and Health Sciences, Norwegian University of Science and Technology (NTNU), 7491 Trondheim, Norway; 4Centre for Obesity and Innovation (ObeCe), Clinic of Surgery, St. Olav’s University Hospital, 7006 Trondheim, Norway; 5Department of Medicine, Møre and Romsdal Hospital Trust, 6026 Ålesund, Norway; 6Nord-Trøndelag Hospital Trust, Clinic of Surgery, Namsos Hospital, 7800 Namsos, Norway; 7Department of Surgery, Møre and Romsdal Hospital Trust, 6026 Ålesund, Norway

**Keywords:** NAFLD, Roux-en-Y gastric bypass, liver fibrosis, FIB-4, NAFLD Fibrosis index, non-invasive fibrosis scores, obesity

## Abstract

Severe obesity is a strong risk factor for non-alcoholic fatty liver disease (NAFLD). Roux-en-Y gastric bypass (RYGB) surgery effectively induces weight loss, but few studies have described the long-term effects of RYGB on NAFLD-related fibrosis. Data from 220 patients with severe obesity operated by RYGB in Central Norway were analysed. Variables incorporated in NAFLD Fibrosis Score (NFS), Fibrosis-4 (FIB-4) index and anthropometric data were collected before surgery and a mean of 11.6 years postoperatively. FIB-4 > 1.3 or NFS > 0.675 were used as cut-off values for advanced fibrosis. Proportions with advanced fibrosis decreased from 24% to 14% assessed by FIB-4 and from 8.6% to 2.3% using NFS, with resolution rates of advanced fibrosis of 42% and 73%, respectively. The shift towards lower fibrosis categories was significant (NFS *p* < 0.0001; FIB-4 *p* = 0.002). NFS decreased from −1.32 (IQR −2.33–−0.39) to −1.71 (IQR −2.49–−0.95, *p* < 0.001) 11.6 years after surgery, whereas FIB-4 did not change: 0.81 (IQR 0.59–1.25) to 0.89 (IQR 0.69–1.16, *p* = 0.556). There were weak correlations between change in fibrosis scores and weight loss. In conclusion, the majority of patients with advanced fibrosis at baseline had improvement after 11.6 years. Factors associated with reduction in fibrosis were not identified.

## 1. Introduction

Non-alcoholic fatty liver disease (NAFLD) is a condition where fat accumulates in the liver. The diagnosis represents a range of different stages of chronic liver disease, including non-alcoholic steatohepatitis (NASH), a more severe form with a significant risk of cirrhosis development. NAFLD is defined by the presence of steatosis in >5% of hepatocytes and requires secondary causes including excessive alcohol consumption (>30 g/d for men, >20 g/d for women) to be ruled out [1]. In the Western world, NAFLD is now the most frequent cause of chronic liver disease [2] with a global prevalence estimated to 25% [3]. A study based on data from the Nord-Trøndelag Health Study (HUNT) using the Fatty Liver Index found a prevalence of 36% [4]. It is predicted that cirrhosis caused by NAFLD will be the most common indication for liver transplantation in several Western countries within 2030 [2], Nordic countries included [5]. NAFLD is associated with overweight and obesity [6], as well as metabolic syndrome and type 2 diabetes mellitus (T2DM) [2]. Clinical symptoms of NAFLD are seldom present before cirrhosis develops. The condition is underdiagnosed since up to 80% of patients with NAFLD also have normal liver enzyme values [7]. Few isolated biochemical parameters have a satisfactory sensitivity and specificity in detecting NAFLD, NASH and fibrosis [8]. However, combining variables into mathematical algorithms provides a higher prediction accuracy. Although histological examination of a liver biopsy is the gold standard for classifying NAFLD, NASH and fibrosis, non-invasive methods such as ultrasound (US), computer tomography (CT) and magnetic resonance imaging (MRI) are increasingly being used [1]. Alternative tools are biomarkers alone or combined with anthropometric variables and the presence of associated diseases. Several scoring systems and biomarker panels have been developed and validated for this purpose [9]. The Fibrosis-4 index (FIB-4) and NAFLD fibrosis score (NFS) have both been compared to US elastography and liver histology [9]. The use of these scores is recommended in patients with low-risk of advanced fibrosis/cirrhosis by the Clinical Practice Guidelines for the management of NAFLD developed by European Association for the Study of the Liver (EASL), European Association for the Study of Diabetes (EASD) and European Association for the Study of Obesity (EASO) [1].

Patients with NAFLD have an increased overall mortality risk, in particular due to cardiovascular disease and liver-related events including hepatocellular carcinoma [10]. The first-line treatment is lifestyle changes. However, weight loss of 7–10% seems necessary to induce regression of NASH [11]. People with severe obesity 18–60 years of age, who are not able to attain this weight loss, may be eligible for bariatric surgery such as Roux-en-Y gastric bypass (RYGB). In addition to long-term weight reduction, RYGB also aims to reduce associated comorbidities, especially metabolic syndrome and T2DM, with a subsequent decreased risk of cardiovascular events and mortality [11], as well as increase quality of life. Several studies have demonstrated short- and mid-term beneficial effects of RYBG on NAFLD [12,13]; however, long-term data are scarce. This study aimed to examine the effect of RYGB on NAFLD fibrosis assessed by FIB-4 and NFS more than a decade after surgery. Furthermore, the study aimed to identify characteristics of patients with resolution of fibrosis.

## 2. Materials and Methods

### 2.1. Study Design

The study was a retrospective analysis of data collected as a part of the Bariatric Surgery Observation Study (BAROBS). BAROBS is a follow-up study of patients operated for severe obesity with RYGB between 2003 and 2009 at three public hospitals in Central Norway Regional Health Authority (Helse Midt-Norge). The objective of the BAROBS study was to evaluate the long-term effects of RYGB on weight loss, comorbidities associated with obesity as well as quality of life and overall health. The patients were invited to a follow-up visit between 2018 and 2020, and 546 patients (58.7%) of the 930 who underwent RYGB in the period participated. Sixteen patients had RYGB as a secondary bariatric procedure, leaving 530 patients with RYGB as their primary bariatric procedure. The data set consists of laboratory tests and anthropometric data collected before surgery and in conjunction with the follow-up visit, in addition to information from the patients’ preoperative medical records, a self-administered questionnaire and an interview with a physician the day of the BAROBS follow-up.

### 2.2. Criteria for Surgery

The criteria for being approved for RYGB surgery were in accordance with the national and international guidelines at the time:

18–60 years of age

Body mass index (BMI) > 40 kg/m^2^ or BMI > 35 kg/m^2^ with obesity related comorbidities, when obesity was not caused by an endocrine disorder

Failed attempts at non-surgical weight loss

No medical or psychological contraindications to surgery, eating disorder, known alcohol or medication abuse or other conditions not compatible with necessary lifestyle changes and follow-up postoperatively.

### 2.3. Exclusion Criteria

Patients were excluded from this particular study if they had:Missing laboratory tests needed to calculate FIB-4 or NFSExcessive alcohol consumption defined as >1 unit/d for women, >2 units/d for men [14] and/or disclosing a problem with alcohol abuse postoperatively.

A flowchart of the study population is presented in Figure 1. A strict threshold for excessive alcohol consumption was chosen, as Rehm et.al. found that already one drink per day was associated with a significantly elevated mortality risk [14]. Patients who have undergone bariatric surgery potentially have an increased long-term risk of alcohol-related cirrhosis [15,16] that in part might be due to altered alcohol metabolism [17]. Based on the considerations above, the stricter threshold was chosen to separate the effects of RYGB *per se* and alcohol intake on fibrosis scores.

### 2.4. Fibrosis Estimation

The NAFLD fibrosis score (NFS) comprises six variables: age, BMI, platelet count, albumin, presence of hyperglycaemia/diabetes and AST/ALT ratio (AAR). Patients were defined as having T2DM if they had been diagnosed with diabetes or had a HbA1c ≥ 48 mmol/mol (≥6.5%), according to the national guidelines by the Norwegian Directorate of Health. The Fibrosis-4 (FIB-4) index is composed of the following variables: platelet count, age and levels of ALT and AST. The EASL Clinical Practice Guidelines on non-invasive tests for evaluation of liver disease severity and prognosis recommend cut-off values for FIB-4 < 1.3 and NFS < −1.455 to rule out advanced fibrosis in patients with NAFLD [18]. A FIB-4 ≥ 1.3 suggests an intermediate to high risk and should prompt further investigation with transient elastography (TE) and/or patented serum tests. A NFS > 0.675 indicates the presence of advanced fibrosis [19] and has a high specificity [18]. Based on these recommendations, we used FIB-4 as a dichotomous variable with a cut-off of 1.3 and NFS as a polychotomous variable divided into three groups: <−1.455 low risk, −1.455 to 0.675 intermediate risk and >0.675 high risk.

### 2.5. Statistical Analysis

Descriptive data are presented as frequency (n (%)) for categorical data and mean ± standard deviation (SD) or median and interquartile range (IQR) for numerical data depending on distribution. Normality was assessed using the Shapiro–Wilk test. The Pearson χ^2^ or Fishers exact test were used for comparisons of independent categorical variables between groups and McNemar’s test for paired categorical variables. Comparisons of independent continuous variables between groups were analysed by independent samples Student’s t test for two variables and one-way ANOVA for more than two variables, both when data were normally distributed. Comparisons of independent non-parametric continuous variables were analysed by Mann–Whitney U test for two variables and Kruskal–Wallis test for more than two variables. The paired Wilcoxon’s signed rank test was used to compare FIB-4 and NFS before and after surgery, while paired Student’s t test was used for comparing paired data of continuous variables that were normally distributed. Correlation between weight loss and changes in fibrosis scores was analysed with Pearson or Spearman tests depending on linearity, outliers and type of data (normal or continuous). *p* values < 0.05 were considered statistically significant. All statistical analyses were conducted using IBM SPSS Statistics version 27.0 (IBM Corporation, Armonk, NY, USA).

## 3. Results

### 3.1. Patient Characteristics

A total of 220 patients, whereof 172 (78.2%) were female, were included in this study. Median BMI prior to RYGB surgery was 43.5 kg/m^2^ (IQR 40.4–46.9), while median BMI at the time of BAROBS was decreased to 33.1 kg/m^2^ (IQR 29.5–38.1, *p* < 0.001) after a mean follow-up time of 11.6 years (SD ± 1.65). Patient characteristics are summarized in Table 1 and Table 2.

### 3.2. Change in NFS and FIB-4 Scores during the Observation Period

The median NFS at baseline was −1.32 (IQR −2.33–−0.39) and decreased to −1.71 (IQR −2.49–−0.95, *p* < 0.001) 11.6 years after RYGB surgery. However, the median FIB-4 score did not change with a median of 0.81 (IQR 0.59–1.25) preoperatively and 0.89 (IQR 0.69–1.16, *p* = 0.556) at follow-up. A comparison of results at baseline and at follow-up is summarized in Table 3.

### 3.3. Reduction of NFS and FIB-4 in High-Risk Patients

Nineteen patients (8.6%) had a preoperative high risk of NAFLD-related fibrosis according to NFS compared to five (2.3%) patients 11.6 years after surgery. There was a significant overall shift towards lower risk categories (*p* < 0.0001) (Figure 2). Three out of the five patients with a high postoperative risk moved from the intermediate- to high-risk group, whereas two patients had a persistent high risk. Fifty-two (24%) patients had an elevated risk of fibrosis according to the FIB-4 index at baseline compared to 31 (14%) patients 11.6 years after surgery. The shift towards lower risk categories was significant (*p* = 0.002). Eleven out of the 31 patients (35%) experienced an increased risk postoperatively, moving from the low- to high-risk group, whereas the remaining 20 (65%) patients had a persistent elevated risk.

### 3.4. Correlation between Changes in NFS and FIB-4, Weight Loss and Remission of Type 2 Diabetes Mellitus

There was a strong correlation between changes in NFS and FIB-4 (r = 0.658, *p* < 0.0001) (Figure 3A). There was no correlation between change in FIB-4 and weight loss variables (%EWL (r = 0.111, *p* = 0.100) and %TWL (r = 0.110, *p* = 0.104)) and only a weak negative correlation between the decrease in NFS and weight loss (%EWL (r = −0.251, *p* < 0.0001) and %TWL (r = −0.280, *p* < 0.0001)). Similarly, there was no correlation between change in FIB-4 (r = −0.092, *p* = 0.172) and change in T2DM status, and the correlation between change in T2DM and change in NFS was weak but significant (r = 0.203, *p* = 0.003).

### 3.5. Characteristics of Patients with Elevated Fibrosis Risk at Baseline and Subsequent Improvement

To assess characteristics of patients who seemed to benefit more from RYGB surgery in terms of fibrosis reduction, we performed a sub-analysis of patients with a high risk of fibrosis (FIB-4 ≥ 1.3) at baseline and compared patients with reduction versus no reduction in fibrosis scores 11.6 years after RYGB surgery. Patients with a fibrosis risk-reduction (FIB-4 < 1.3) and the group with persistent elevated risk (FIB-4 ≥ 1.3) did not differ in terms of age, sex, BMI, weight loss or T2DM prevalence (Table 4). An equivalent analysis of the patients with a baseline NFS > 0.675 was omitted due to small sample size (16 vs. 3 patients).

We conducted a similar analysis of the whole population (n = 220), comparing patients with a reduction in fibrosis scores with patients with an increase, using scores as continuous variables. According to FIB-4, patients with an increased score at follow-up had a greater weight loss with %EWL 57.8 ± 26.1 vs. 50.2 ± 28.0 (*p* = 0.039) and %TWL 24.4 ± 10.6 vs. 20.7 ± 11.6 (*p* = 0.016) in the group with an increased score versus the group with a decreased score, respectively (Table A1 in Appendix A). The patients with an increased NFS at follow-up experienced less weight loss with %EWL 49.2 ± 28.5 vs. 57.6 ± 25.9 (*p* = 0.026) and %TWL 20.2 ± 11.5 vs. 24.3 ± 10.7 (*p* = 0.008) in the group with an increased score versus the group with a decreased score, respectively (Appendix A, Table A1).

## 4. Discussion

We found that RYGB surgery led to a significant reduction of liver fibrosis risk assessed by the non-invasive FIB-4 index and NFS 11.6 years postoperatively, although the improvement of FIB-4 was significant only when assessed as a dichotomous variable. The proportion of patients with advanced fibrosis was reduced from 24% to 14% assessed by FIB-4 and from 8.6% to 2.3% assessed by NFS, implying resolution rates for advanced fibrosis of 42% and 73%, respectively. The above-mentioned reductions in fibrosis scores were observed despite age being a component in both scores and a mean 11.6-year follow-up time. The NFS and FIB-4 index were developed in studies where most patients were between the age of 35 and 65 [20]. Age has later been identified as a confounding factor for NFS and FIB-4 in diagnosing advanced fibrosis since the scores overestimate the risk of fibrosis in individuals >65 years of age [20]. A higher cut-off for ruling out advanced fibrosis in patients >65 years has therefore been proposed (2.0 for FIB-4 and 0.12 for NFS); however, further validation studies seem necessary before age-dependent cut-off values can be used in clinical practice. For patients ≤ 35 years, the overall performance of both scores is also poor [20]. In our study population, 74 vs. 5 patients were ≤35 years and 0 vs. 20 patients were >60 years at baseline vs. 11.6 years later, respectively. The prevalence of biopsy-proven fibrosis in obese patients has been reported in the range of 25–47%, also including patients undergoing bariatric surgery [21,22,23,24]. In the current study, the prevalence of fibrosis assessed by NFS and FIB-4 was found to be lower (8.6–24%). A high proportion of young patients at baseline might contribute to a lower prevalence of advanced fibrosis, particularly in women where the risk of fibrosis increases after menopause onset [25]. Two recent meta-analyses have reported that RYGB had a positive effect on NAFLD after short- to mid-term follow-up time of 1–60 months, with improvement or resolution of histological steatosis and steatohepatitis in the majority of patients, as well as resolution of fibrosis in 28–51% [12,13]. Importantly, studies assessing resolution of liver fibrosis after RYGB with NFS and/or FIB-4 have found improvement in agreement with studies using biopsies [26,27,28,29]. The proportion of patients with a low postoperative fibrosis score was similar in our population to that in studies with shorter follow-up times [11,12,29,30,31,32,33]. The resolution rates of advanced fibrosis evaluated by NFS have been reported to be 55% [29] and 100%, although in a relatively small study [34], both one year after RYGB. Others have also reported similar proportions of improvement in NFS after one year [35]. However, our findings indicate that the beneficial effects of RYGB on fibrosis persist long term for 11.6 years after surgery.

The exact mechanisms behind improvement of NAFLD-related fibrosis are uncertain. There were only weak (NFS) or non-significant (FIB-4) correlations between change in fibrosis scores and weight loss variables and reduced prevalence of T2DM. Other predictors of improved fibrosis scores could not be identified. Patients with improved FIB-4 actually experienced less weight loss (%TWL and %EWL), while patients with improved NFS experienced greater weight loss, although the latter association should be interpreted with care, since BMI is a component of NFS. This indicates that although weight loss *per se* is important in NAFLD treatment, other pathophysiological mechanisms play a role in fibrosis regression, which is also suggested by two other patient studies [36,37]. These clinical observations are supported by experimental animal studies demonstrating that improvement of glycemia [38] and reversal of NAFLD after RYGB occurs independent of weight loss [39]. We did not find factors associated with improvement of fibrosis. Perhaps of equal importance is that we did not identify factors associated with lack of improvement or progression. In a clinical setting, this implies that RYGB could be considered in all patients with advanced fibrosis, and RYGB seems to be beneficial and relatively safe in patients with cirrhosis [40].

Strengths of the study include the long follow-up time of a relatively large study population. Limitations of the study include the lack of liver biopsy as a gold standard for evaluation of fibrosis. The study was retrospective, and fibrosis scores could be analysed only in patients with complete data for all score components as illustrated in Figure 1. Furthermore, laboratory tests were taken at different time points within the year before surgery. Various medications used during the study period could potentially affect non-invasive fibrosis scores, and such effects could not be delineated. Systematic screening for other liver diseases was not performed; however, the prevalence of undiagnosed chronic liver diseases such as viral hepatitis [41] in the study population is likely to be low. We could not identify factors associated with improvement of liver fibrosis, and our finding should be validated in other patient cohorts.

## 5. Conclusions

This study of a RYGB cohort with a unique long-term follow-up of 11.6 years suggests that RYGB leads to resolution of NAFLD fibrosis in the majority of patients with high risk. Improvement of NAFLD and related fibrosis may be among the major benefits of RYGB, and there was no correlation with degree of weight loss or remission of T2DM. Therefore, we have not been able to identify any specific groups of patients who seem to benefit more from bariatric surgery. Considering the increasing rate of obesity globally, studies of NAFLD and fibrosis seem essential.

## Figures and Tables

**Figure 1 jcm-11-04910-f001:**
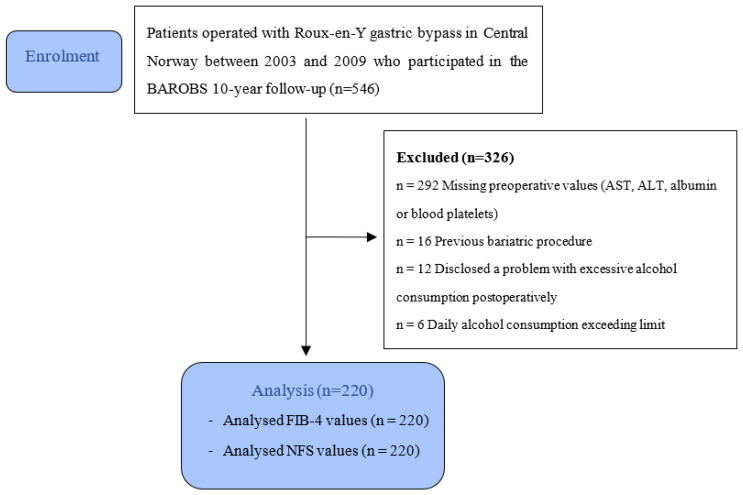
Flow-chart of the study population.

**Figure 2 jcm-11-04910-f002:**
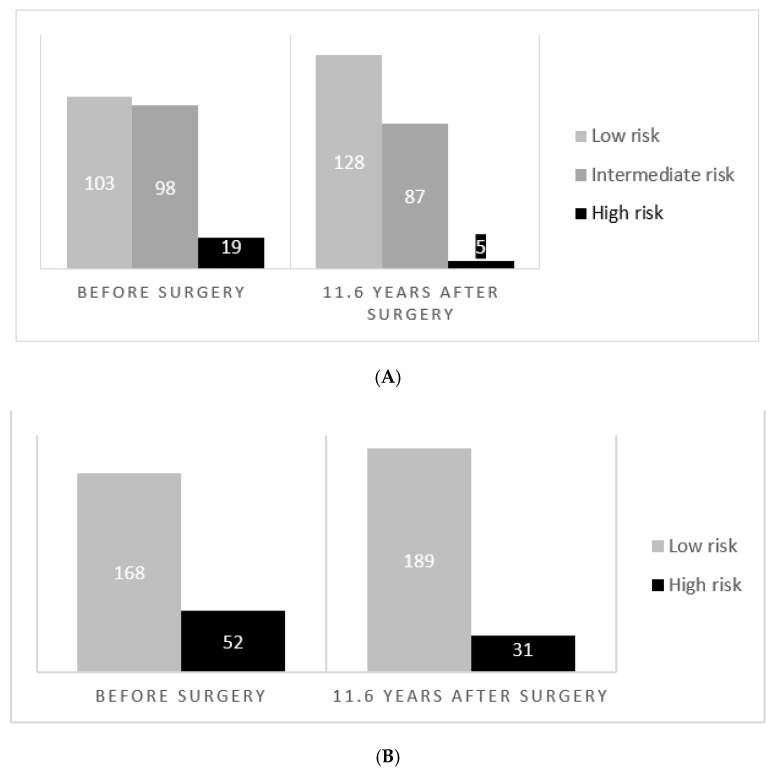
Risk distribution according to NAFLD fibrosis Score (**A**) and Fibrosis-4 index (**B**) before and 11.6 years after Roux-en-Y gastric bypass.

**Figure 3 jcm-11-04910-f003:**
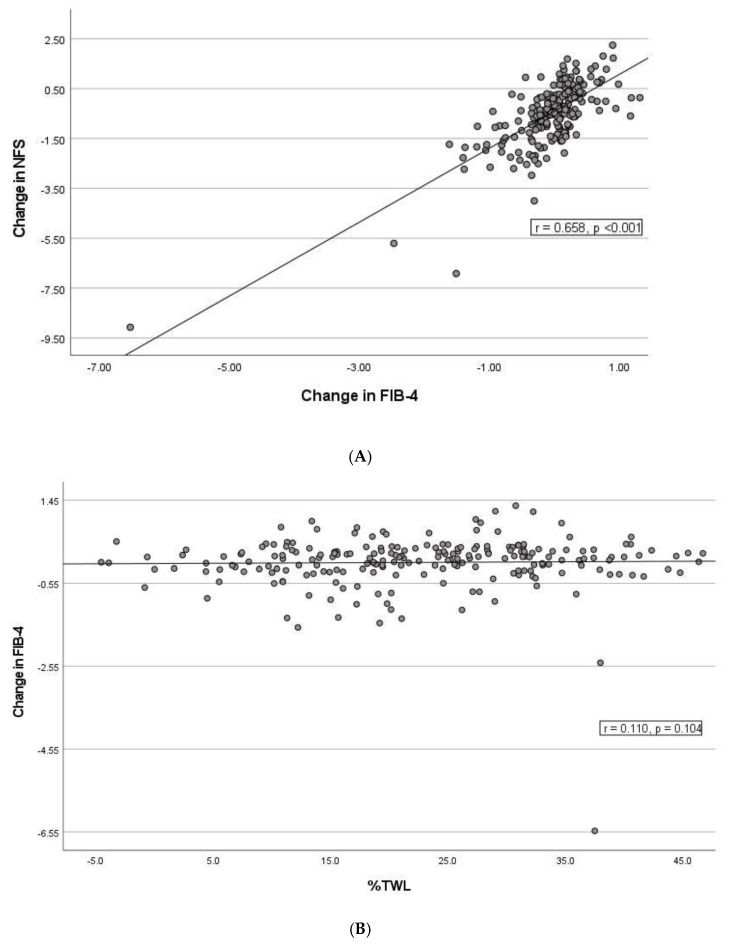
Scatterplots illustrating the correlations between change in NAFLD Fibrosis Score (NFS) and FIB-4 index (**A**) and between %total weight loss and FIB-4 (**B**) and NFS (**C**) following Roux-en-Y gastric bypass surgery. FIB-4, Fibrosis-4 index; NFS, non-alcoholic fatty liver disease fibrosis score; %TWL, percentage total weight loss.

**Table 1 jcm-11-04910-t001:** Study population prior to Roux-en-Y gastric bypass surgery.

Variable	Overall (n = 220)	FIB-4		NFS		
Characteristics		<1.3 (n = 168)	≥1.3 (n = 52)	<−1.455 (n = 100)	Intermediate (n = 101)	>0.675 (n = 19)
Age, years, mean ± SD	40.1 ± 9.5	37.7 ± 8.4	47.6 ± 9.0 ***^,a^	36.8 ± 7.8	42.0 ± 9.9	46.6 ± 9.4 ***^,b^
Female, n (%)	172 (78.2)	135 (80.4)	37 (71.2) ^a^	79 (79.0)	77 (76.2)	16 (84.2) ^b^
BMI, kg/m^2^*,* median (IQR)	43.5 (40.4–46.9)	43.7 (40.6–47.4)	43.2 (39.5–45.7) ^a^	42.7 (40.3–45.5)	43.8 (40.6–47.4)	45.7 (41.2–51.2) *^,b^
T2DM n (%)	40 (18.2)	28 (16.7)	12 (23.1) ^a^	8 (8.0)	27 (26.7)	5 (26.3) **^,b^
Diet controlled	4 (1.8)	3 (1.8)	1 (1.9)	0 (0)	3 (3.0)	1 (5.3)
Oral diabetics	23 (10.5)	14 (8.3)	9 (17.3)	6 (6.0)	14 (13.9)	3 (15.8)
Insulin	11 (5.0)	9 (5.4)	2 (3.8)	2 (2.0)	8 (7.9)	1 (5.3)
AST, U/L, median (IQR)	37.5 (28.0–49.8)	34.0 (27.0–44.0)	53.5 (42.0–71.3) ***^,a^	36.0 (27.0–46.0)	38.0 (29.5–47.5)	54.0 (37.0–69.0) **^,b^
ALT, U/L, median (IQR)	34.0 (23.3–50.8)	35.0 (24.5–54.0)	30.5 (19.0–44.8) ^a^	38.5 (29.0–60.8)	32.0 (23.0–46.0)	20.0 (18.0–34.0) ***^,b^
PLT, ×10^9^/L, median (IQR)	297 (253–349)	311 (268–359)	248 (210–292) ***^,a^	341 (301–384)	271 (239–301)	225 (188–286) ***^,b^
ALB, g/L, median (IQR)	42.0 (40.0–44.0)	42.0 (40.0–44.0)	42.0 (40.0–44.8) ^a^	43.0 (41.0–45.0)	41.0 (39.5–43.5)	41.0 (39.0–43.0) **^,b^
FIB-4, median (IQR)	0.81 (0.59–1.25)	0.72 (0.55–0.89)	1.61 (1.46–2.07) ***^,a^	0.62 (0.47–0.79)	0.96 (0.77–1.40)	2.08 (1.56–2.99) ***^,b^
NFS, median (IQR)	−1.32 (−2.33–−0.39)	−1.74 (−2.51–−1.03)	0.07 (−0.54–0.94) ***^,a^	−2.40 (−2.81–−1.89)	−0.61 (−1.14–−0.07)	1.17 (0.91–1.71) ***^,b^

^a^ Comparison between FIB-4 categories; ^b^ Comparison between NFS categories; * *p* < 0.05; ** *p* < 0.01; *** *p* < 0.001; FIB-4, Fibrosis-4 index; NFS, non-alcoholic fatty liver disease fibrosis score; BMI, Body Mass Index; T2DM, type 2 diabetes mellitus; AST, aspartate aminotransferase; ALT, alanine transaminase; PLT, blood platelets; ALB, albumin; IQR, interquartile range; SD, standard deviation.

**Table 2 jcm-11-04910-t002:** Study population 11.6 years after Roux-en-Y gastric bypass surgery.

Variable	Overall (n = 220)	FIB-4		NFS		
Characteristics		<1.3 (n = 189)	≥1.3 (n = 31)	<−1.455 (n = 128)	Intermediate (n = 87)	>0.675 (n = 5)
Age, years, mean ± SD	51.7 ± 9.6	50.2 ± 9.0	60.5 ± 8.6 ***^,a^	48.8 ± 8.7	55.0 ± 9.3	65.1 ± 10.4 ^b^
Female n (%)	172 (78.2)	152 (80.4)	20 (64.5) *^,a^	108 (84.4)	60 (69.0)	4 (80.0) *^,b^
BMI, kg/m^2^, median (IQR)	33.1 (29.5–38.1)	33.1 (29.6–38.4)	32.9 (29.0–36.1) ^a^	31.7 (28.5–35.3)	35.8 (31.9–41.0)	36.7 (29.9–49.2) ***^,b^
BMI nadir, kg/m^2^, median (IQR)	28.0 (26.0–31.0)	28.0 (26.0–32.0)	28.0 (26.0–30.0) ^a^	27.0 (24.3–29.0)	29.0 (27.0–33.0)	35.0 (30.5–42.0) ***^,b^
%TWL, mean ± SD	22.8 ± 11.2	22.4 ± 11.4	25.2 ± 9.32 ^a^	25.5 ± 10.6	19.0 ± 11.1	19.3 ± 10.3 ***^,b^
%EWL, mean ± SD	54.5 ± 27.1	53.2 ± 27.5	62.0 ± 24.0 ^a^	61.9 ± 26.1	44.0 ± 25.2	47.1 ± 28.2 ***^,b^
T2DM n (%)	23 (10.5)	18 (9.5)	5 (16.1) ^a^	3 (2.3)	17 (19.5)	3 (60.0) ***^,b^
AST, U/L, median (IQR)	21.0 (17.0–25.0)	20.0 (17.0–24.0)	26.0 (23.0–31.0) ***^,a^	21.0 (17.0–25.0)	20.0 (18.0–25.0)	21.0 (17.5–31.0) ^b^
ALT, U/L, median (IQR)	21.0 (17.0–27.0)	21.0 (17.0–27.0)	22.0 (16.0–31.0) ^a^	21.0 (17.0–27.8)	20.0 (16.0–28.0)	21.0 (19.5–25.0) ^b^
PLT, ×10^9^/L, median (IQR)	252 (220–301)	265 (225–307)	188 (175–231) ***^,a^	283 (239–318)	229 (199–252)	176 (149–209) ***^,b^
ALB, g/L, median (IQR)	44.0 (42.0–45.0)	44.0 (42.0–45.0)	44.0 (42.0–45.0) ^a^	44.0 (43.0–45.8)	43.0 (42.0–45.0)	41.0 (39.5–42.5) **^,b^
FIB-4, median (IQR)	0.89 (0.69–1.16)	0.81 (0.66–1.06)	1.59 (1.48–1.81) ***^,a^	0.77 (0.60–0.93)	1.14 (0.89–1.35)	2.32 (1.04–2.69) ***^,b^
NFS, median (IQR)	−1.71 (−2.49–−0.95)	−1.83 (−2.65–−1.23)	−0.50 (−0.99–0.10) ***^,a^	−2.31 (−2.89–−1.83)	−0.91 (−1.17–−0.50)	1.07 (0.85–1.40) ***^,b^

^a^ Comparison between FIB-4 categories; ^b^ Comparison between NFS categories; * *p* < 0.05; ** *p* < 0.01; *** *p* < 0.001; FIB-4, Fibrosis-4 index; NFS, non-alcoholic fatty liver disease fibrosis score; BMI, body mass index; %TWL, percentage total weight loss; %EWL, percentage excess weight loss; T2DM, type 2 diabetes mellitus; AST, aspartate aminotransferase; ALT, alanine transaminase; PLT, blood platelets; ALB, albumin; IQR, interquartile range; SD = standard deviation.

**Table 3 jcm-11-04910-t003:** Comparison of NFS, FIB-4 and their components before and 11.6 years after Roux-en-Y gastric bypass surgery.

Characteristics (n = 220)	Before Surgery	11.6 Years after Surgery	*p* Value
NFS, median (IQR)	−1.32 (−2.33–−0.39)	−1.71 (−2.49–−0.95)	<0.001
FIB-4, median (IQR)	0.81 (0.59–1.25)	0.89 (0.69–1.16)	0.556
BMI, kg/m^2^, median (IQR)	43.5 (40.4–46.9)	33.1 (29.5–38.1)	<0.001
T2DM n (%)	40 (18.2)	23 (10.5)	<0.001
AST, U/L, median (IQR)	37.5 (28.0–49.8)	21.0 (17.0–25.0)	<0.001
ALT, U/L, median (IQR)	34.0 (23.3–50.8)	21.0 (17.0–27.0)	<0.001
PLT, ×10^9^/L, median (IQR)	297 (253–349)	252 (220–301)	<0.001
ALB, g/L, median (IQR)	42.0 (40.0–44.0)	44.0 (42.0–45.0)	<0.001

NFS, non-alcoholic fatty liver disease fibrosis score; FIB-4, Fibrosis-4 index; BMI, body mass index; T2DM, type 2 diabetes mellitus; AST, aspartate aminotransferase; ALT, alanine transaminase; PLT, blood platelets; ALB, albumin; IQR, interquartile range; SD, standard deviation.

**Table 4 jcm-11-04910-t004:** Patients with FIB-4 ≥ 1.3 at baseline with improvement in FIB-4 index vs. patients without improvement.

Variable	FIB-4 < 1.3 (n = 32)	FIB-4 ≥ 1.3 (n = 20)	*p* Value
Age baseline, years, mean ± SD	45.9 ± 9.33	50.2 ± 7.84	0.099
Female n (%)	25 (78.1)	12 (60.0)	0.213
BMI baseline, kg/m^2^, median (IQR)	43.1 (39.5–45.8)	44.1 (39.3–45.6)	0.821
BMI follow-up, kg/m^2^, median (IQR)	33.1 (28.8–37.0)	31.9 (28.2–36.6)	0.529
BMI nadir, kg/m^2^, median (IQR)	28.0 (25.0–32.0)	28.0 (25.5–33.0)	0.741
%TWL, mean ± SD	21.7 ± 11.8	26.8 ± 9.55	0.108
%EWL, mean ± SD	53.1 ± 28.7	66.1 ± 24.9	0.101
T2DM baseline, n (%)	9 (28.1)	3 (15.0)	0.330
T2DM follow-up, n (%)	6 (18.8)	2 (10.0)	0.463

FIB-4, Fibrosis-4 index; BMI, body mass index; %TWL, percentage total weight loss; %EWL, percentage excess weight loss; T2DM, type 2 diabetes mellitus; IQR, interquartile range; SD, standard deviation.

## Data Availability

Due to ethical and legal regulations of clinical research, the data cannot be shared.

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
