# Peer review of "The Effect of Roux-en-Y Gastric Bypass on Non-Alcoholic Fatty Liver Disease Fibrosis Assessed by FIB-4 and NFS Scores—An 11.6-Year Follow-Up Study"

_jcm, 2022, doi:10.3390/jcm11164910_

Round 1
Reviewer 1 Report
Sandvik et al. did a great job in their manuscript describing a retrospective analysis of patient’s fibrosis score before and after gastric bypass using two non-invasive scoring systems adopted by the liver societies the NFS and the FIB-4. The results show significant reduction in fibrosis in those with advanced disease or high risk NAFLD using both scores and overall reduction using the NFS only. there were no patient-specific factors that were associated with this reduction.
Overall the study was well prepared, the hypothesis, design, data collection and analysis were all very good, and the manuscript was well written. I have no major concerns about it but I have these following suggestion for the authors :
1- I would add more to the limitations of the study reasons of why there were no factors associated with fibrosis reduction. Was it population, study design, etc.
2- I would also comment on the clinical value of this finding and how it can be used for example in patients’ wit high vs low risk NALFD.
3- Finally, it would be nice to change Figure 2 to a form that shows the effect of reduction over time.
Author Response
Sandvik et al. did a great job in their manuscript describing a retrospective analysis of patient’s fibrosis score before and after gastric bypass using two non-invasive scoring systems adopted by the liver societies the NFS and the FIB-4. The results show significant reduction in fibrosis in those with advanced disease or high risk NAFLD using both scores and overall reduction using the NFS only. there were no patient-specific factors that were associated with this reduction.
Overall the study was well prepared, the hypothesis, design, data collection and analysis were all very good, and the manuscript was well written. I have no major concerns about it but I have these following suggestion for the authors:
- I would add more to the limitations of the study reasons of why there were no factors associated with fibrosis reduction. Was it population, study design, etc.
Response: As requested, we have added to the limitations of the study that we only used one cohort and that the analyses should be repeated in different cohorts. Identifying factors associated with fibrosis reduction is indeed very interesting. Such factors could be of clinical relevance as well as be suggestive of mechanisms behind fibrosis reduction beyond weight loss per se. However, our data and study design did not allow further exploration of this question.
- I would also comment on the clinical value of this finding and how it can be used for example in patients’ wit high vs low risk NALFD.
Response: The clinical value of this findings (no factors associated with fibrosis improvement, improvement for most patients with advanced fibrosis) could be rephrased as we did not identify factors associated with no improvement or progression. RYGB could therefore be considered in all patients with advanced fibrosis.
The cohort consisted of patients with severe obesity and a high risk of NAFLD and can not be used in patients with low risk of NAFLD.
- Finally, it would be nice to change Figure 2 to a form that shows the effect of reduction over time.
Response: Figure 2 illustrates the change in FIB-4 and NFS at the two different time points. The effect of reduction over time could perhaps be more evident with a different design and we have changed the graph to a clustered columns and it may the reductions in higher risk categories then become more visible.
Reviewer 2 Report
this is very interesting article however I think it is better to include more articles with same methods in discussion about liver fibrosis improvement before and after surgery .
thanks
Author Response
T
This is very interesting article however I think it is better to include more articles with same methods in discussion about liver fibrosis improvement before and after surgery.
Response: We thank the reviewer for pointing this out. In addition to the studies already referred to in the original manuscript, we have added two more references (34 and 35) and mentioned these findings in the discussion. In fact, the number of studies that has used FIB-4 and NFS as non-invasive scores before and after RYGB seems low, the studies are short term and have a focus on other parameters when presenting their analyses.